# OpenReview forum: "Symmetry-Robust 3D Orientation Estimation"
_ICML.cc/2025/Conference — ICML 2025 poster_

### Official Review · Reviewer_ffTb · 2025-02-18

**Overall Recommendation:** 3

**Summary:**

This work presents a full orientation estimation method for generic shapes. Concretely, a two-stage framework is proposed for this task. The method first uses a quotient orienter to recover the shape's orientation up to octahedral symmetries by continuous regression. Then a flipper is employed to predict one of 24 octahedral flips that returns the first-stage output to canonical orientation via standard classification. Additionally, a conformal prediction stage is used to enable the flipper to output adaptive prediction sets, resolving ambiguities in the results through human-in-the-loop interaction. Experimental results show that the proposed method achieves SOTA performance in up-axis prediction and full-orientation recovery.

## update after rebuttal
The reviewer would be happy to accept this paper if the title is changed to "3D Orientation Estimation for Symmetric Shapes" as the reviewer `gQb9` suggested.

**Claims And Evidence:**

Mostly yes. But the claim of "anything" is somewhat problematic.

**Essential References Not Discussed:**

No.

**Experimental Designs Or Analyses:**

Yes. There could be some more experimental analyses to support the claim of "anything".

**Methods And Evaluation Criteria:**

Yes.

**Other Comments Or Suggestions:**

- The initial two sentences of both the Abstract and the Introduction are essentially identical, rendering them somewhat redundant.
- The titles of certain subsections should not end with full stops.
- How were the standard deviations in Table 2 calculated?

**Other Strengths And Weaknesses:**

Strengths
- As noted by the authors, this is the first attempt to solve the task of full-orientation estimation for generic shapes without class information.
- The paper analyzed why a naive regression approach would fail for full orientation estimation for generic shapes.
- The code is available.

Weaknesses
- In the experiments, the ShapeNet dataset is divided into a 90% training split and a 10% testing split. Then, where does the validation set come from?
- How robust is the method to noise? In practice, reconstructed shapes frequently include noise points. Could the authors evaluate the method’s resilience to noisy shapes, especially nearly symmetric shapes?
- The term "anything" should be used very carefully. To show that the method can achieve the "anything" capability, more experimental results might be needed. For instance, the method can be tested on human/face/hand/animal shapes and protein molecule shapes.
- Could the authors provide runtime analyses for both the training and inference stages of the method?

**Questions For Authors:**

See above.

**Relation To Broader Scientific Literature:**

This paper is related to orientation estimation, shape canonicalization and geometric deep learning. The main contribution is to make the full orientation estimation possible for a wide range of generic shapes.

**Theoretical Claims:**

Yes, but I am not able to check the correctness of all the proofs.

---

> ### Author Rebuttal · Authors · 2025-03-31
>
> Thank you for your thoughtful review and your helpful comments. In this rebuttal, we will address many of the questions you have asked in your review.
>
> *In the experiments, the ShapeNet dataset is divided into a 90% training split and a 10% testing split. Then, where does the validation set come from?*
>
> Because our two models are relatively costly to train (see below), we did not tune our DGCNN’s hyperparameters and instead used the defaults from the original DGCNN implementation. We chose to subsample 2k points per point cloud in each training iteration to roughly match Upright-Net’s 2048 points, and chose the largest batch sizes possible given our GPU’s memory constraints. We did monitor validation metrics during training, but our decision to end our training runs were driven primarily by computational constraints, and our validation metrics were continuing to improve (albeit very slowly) when we ended our final training runs.
>
> We consequently do not have a separate test set. We have referred to our metrics as “validation” metrics rather than “test” metrics elsewhere in the manuscript, and would be happy to refer to the 90-10 split as a training-validation split. Our results on ModelNet40 and on Objaverse provide a comprehensive picture of our method’s performance on fully unseen data, and our ModelNet40 results show that our method also strongly outperforms Upright-Net on fully unseen data.
>
> *How robust is the method to noise? In practice, reconstructed shapes frequently include noise points. Could the authors evaluate the method’s resilience to noisy shapes, especially nearly symmetric shapes?*
>
> Thank you for this suggestion. We have repeated our experiments testing our pipeline’s up-axis estimation accuracy on Shapenet, where we now add Gaussian noise with a standard deviation of 0.05 and 0.1 to the point clouds (which have been normalized to lie in the unit ball) and normals (we re-normalize the normals after adding noise). We depict the result of this experiment [here](https://imgur.com/a/K1rTayu). Our method is fairly robust to small amounts of noise, with a noise std of 0.05 resulting in our pipeline’s accuracy (\% of shapes with angular error of up-axis prediction $<10^\circ$) dropping from 89.2% to 86.2%. Increasing the noise std results in a larger accuracy penalty, with our pipeline’s accuracy dropping further to 77.6%.
>
> We also highlight that Figure 11 in the appendices depicts our pipeline’s outputs on meshes from Objaverse. These meshes are both highly ood for our pipeline, which was trained on Shapenet, and generally lower-quality than those found in Shapenet.
>
> *Could the authors provide runtime analyses for both the training and inference stages of the method?*
>
> Please see our response to Reviewer 3giS for our method’s inference times. Our orienter takes 420 seconds per epoch to train on a single V100 GPU – so 1719 epochs of training is 9.33 days for the orienter. Our flipper takes 410 seconds per epoch on the same V100 GPU – so 3719 epochs is 17.6 days for the flipper. We found that both models’ validation metrics continued to improve after many epochs of training. Improving our problem’s conditioning so that both models require fewer epochs to converge would be a valuable direction for future work.
>
> *How were the standard deviations in Table 2 calculated?*
>
> The mean and standard deviation of the chamfer distances were computed over the Shapenet validation set.
>
> *The term "anything" should be used very carefully. To show that the method can achieve the "anything" capability, more experimental results might be needed. For instance, the method can be tested on human/face/hand/animal shapes and protein molecule shapes.*
>
> We would be pleased to change our paper's title if the program chairs allow it, especially since we have become aware of another recent paper titled "Orient Anything: Learning Robust Object Orientation Estimation from Rendering 3D Models." For example, we could call our paper "Robust 3D Orientation Estimation for Symmetric Shapes".

---

> > ### Comment · Reviewer_ffTb · 2025-04-02
> >
> > The reviewer appreciates the authors' rebuttal and acknowledges that the newly proposed title is a significant improvement over the original, which sounded too overclaimed. However, to demonstrate that the method is really "robust", the reviewer suggests evaluating the method on real-world reconstructed 3D shapes, such as those obtained from 3DGS, SfM, or SLAM. Relying solely on ShapeNet models with synthetic noise and the relatively noise-free Objaverse shapes is insufficient to demonstrate true robustness. Additionally, the reviewer is unsure whether ICML allows title changes after submission, though this has been permitted in similar conferences like ICLR.

---

### Official Review · Reviewer_yNUC · 2025-02-24

**Overall Recommendation:** 3

**Summary:**

This paper explores the challenges faced by current baselines that aim at predicting 3D-shape orientation. It shows that trying to minimize an L2 distance cannot recover the ground-truth orientation in the presence of intrinsic symmetries as the solution to the L2 distance will not be unique. To address this challenge, the paper proposes a two stage pipeline that(1) predicts a set of approxiamations to the ground truth orientation (which recovers a shape’s orientation up to octahedral symmetries), classify this approximations to return the best orientation.

**Claims And Evidence:**

Look okay. The paper proposes to separately predict approximations to the grounth-truth approximations which they do using their proposed "quotient orienter", and then classify these proposed approximations to return the best approximation using their proposed "flipper". The main idea being that directly solving the regression problem to predict the orientation in the presence of intrinsic symmetries is not possible which they show using their propositions 1-3.

**Essential References Not Discussed:**

No recommendations.

**Experimental Designs Or Analyses:**

Looks okay. However, please refer to the "Methods And Evaluation Criteria" Section.

**Methods And Evaluation Criteria:**

Looks okay. But we have two questions:
- We are wondering in lines 296-306 of the experimental setup. It seems the proposed method is trained on 10K points per shape and 2K points are sampled out of the 10k points per shape for each epoch, while UprightNet is only trained on 2k points in total per shape?
- We would also like to ask the authors how the proposed pipeline performs when trained on a similar dataset to that proposed in Upright-Net?

**Other Comments Or Suggestions:**

No recommendations.

**Other Strengths And Weaknesses:**

No recommendations.

**Questions For Authors:**

Please refer to the "Methods And Evaluation Criteria" Section.

**Relation To Broader Scientific Literature:**

Looks okay.

**Theoretical Claims:**

Propositions 3.1-3.3 all look okay in the paper and appendix.

---

> ### Author Rebuttal · Authors · 2025-03-31
>
> Thank you for your thoughtful review and your helpful comments. In this rebuttal, we will answer the two questions you have posed to us in your review.
>
> *We are wondering in lines 296-306 of the experimental setup. It seems the proposed method is trained on 10K points per shape and 2K points are sampled out of the 10k points per shape for each epoch, while UprightNet is only trained on 2k points in total per shape?*
>
> This is correct. We followed the original implementation of Upright-Net as closely as possible, which calls for drawing a fixed reservoir of 2048 points to use in each iteration. Because one can draw arbitrarily large point clouds from the surface of a mesh, we saw no reason to restrict ourselves to a fixed reservoir of 2048 samples when training our pipeline.
>
> *We would also like to ask the authors how the proposed pipeline performs when trained on a similar dataset to that proposed in Upright-Net?*
>
> Because our models are relatively costly to train, we have not had the opportunity to run this experiment during the rebuttal period. However, we believe that training both methods on all of Shapenet provides a more complete and realistic picture of how each method will perform in practice than training both methods on a small subset of classes from ModelNet40.

---

### Official Review · Reviewer_gQb9 · 2025-03-12

**Overall Recommendation:** 4

**Summary:**

The paper presents a two-stage deep learning method to orient shapes, dubbed Orient Anything. It proves why naive L2 regression of the orientation matrix fails for shapes with symmetries, and presents a theoretical framework to overcome the problem. The method consists in selecting a finite group $\hat{R}$ that contains the rotational symmetries of the object. In practice, the authors choose the octahedral group that contains the 24 rotational symmetries of a cube and captures most of the rotational symmetries of real-world shapes. Then, the problem of orientation estimation is divided into two stages, corresponding to two neural networks: the first regresses the orientation up to a rotation in $\hat{R}$ (so called "quotient regressor"), the second one selects one of the rotations in $\hat{R}$ (so called "flipper"). The composition of the two transformations gives the estimated orientation. Finally, since the orientation of some shapes is inherently ambiguous, the authors propose to use conformal prediction when a human is in the loop, to let the network output adaptive prediction sets whose size varies with the flipper's uncertainty. The method is trained on ShapeNet and quantitatively tested on ShapeNet and ModelNet40, both standard and challenging benchmarks for up-axis estimation, i.e. estimating only the vertical axis of the object, and estimation of the full orientation. It outperforms an existing baseline, UpRightNet, for up-axis estimation and it outperforms some made-up baselines, based again on UpRightNet, for the full orientation case. Some qualitative results of generalization capabilities to ObjaVerse are reported in the appendix.

## Update after rebuttal
I carefully read all reviews and responses. The authors clarified how TTA is implemented and performed an ablation study on its contribution, which was found to be not critical for the overall performance of the method and its superior performance with respect to the baseline. The explanation of the first step of Prop 3.1 is also clear now. I'd suggest to add this explanation in the appendix. The other reviews do not uncover critical weaknesses and the responses to them seems convincing. For all these reasons, I confirm my overall recommendation. However, I agree that the title should be toned down, if possible. In particular, I'm in favor of the proposal by the authors "3D Orientation Estimation for Symmetric Shapes" (dropping "Robust" for the reasons discussed by reviewer ffTb).

**Claims And Evidence:**

The claims are fully supported by clear and convincing theoretical results and experimental evidence.

**Essential References Not Discussed:**

None

**Experimental Designs Or Analyses:**

Experimental designs and analyses are in general sound and valid.

One concern I have is on the effect of test-time augmentation: the authors did not ablate its contribution, so I think OrientAnything without TTA should be added to Table 1 and 2, to assess its importance. Moreover, for the flipper, it is not clear how a single output is selected when not using conformal prediction. The paper just reads "(4) output the plurality prediction", but there are multiple predictions from the multiple augmentations. Moreover, it is not clear how this works when using adaptive sets: in which order are logits from the different shapes ranked? A full rank across different input orientations or is some criterion deployed to choose one of them? I'd suggest to provide more details on TTA.

**Methods And Evaluation Criteria:**

Evaluation criteria make sense and, for the up-axis estimation, are the one used in previous work. The main weakness is the lack of proper baselines for the full orientation case. The baselines created by the authors, being based on UpRightNet, were clearly at a disadvantage, so, in some sense, the second experiment does not provide new evidence in terms of comparisons, Yet, the authors tried their best and the experiment provides the absolute performance of the method on the harder problem of full orientation estimation. Hence, I think it is valuable and should stay in the final version.

**Other Comments Or Suggestions:**

Fonts in Figure 2 are too small.

Figure 4 does not really illustrate that Equation 2 has many solutions. This became clear to me only after reading A.3. I'd suggest to remove Figure 4 and move A.3 in the main text or to create a more informative figure.

line 353 "validation" -> is it "test"?

**Other Strengths And Weaknesses:**

S1. The paper is clearly written and presented.

S2. The idea of dividing orientation estimation into two steps based on the quotient regression is original, sound, non-trivial and effective.

S3. The method is shown to outperform the considered baseline.

W1. Test-time augmentation is not explained clearly, its hyperparameters are not reported and its importance is not ablated.

**Questions For Authors:**

Q1. Please clarify how TTA is realized and its impact on the model performance.

**Relation To Broader Scientific Literature:**

The paper cites the relevant literature and provides a clear original improvement with respect to it, both at the theoretical and at the performance level.

**Theoretical Claims:**

I  believe the theoretical claims are correct. My only issue is with the first step of the proof of proposition 3.1 in A.2. It is not clear why the equation at line 576 is equivalent to Equation 2. In equation 2, the Expected value is on $R \in U(SO(3))$. Not sure why here it becomes the expected values on $R'$ such that $RS = R'S$, even after reading the introduction to the section. I'd suggest to clarify this step.

---

> ### Author Rebuttal · Authors · 2025-03-31
>
> Thank you for your thoughtful review and your helpful comments. We will gladly fix the typos you have pointed out and make the changes to the figures that you have requested in the camera-ready. **Please see our response to Reviewer 3giS for links to new versions of Tables 1 and 2 and Figure 6, where we have now included our method’s performance without TTA.** Below, we provide further details on how we implement TTA, and we clarify the first step of the proof of Prop. 3.1.
>
> **TTA implementation details.**
>
> **Quotient orienter.**
>
> Our use of TTA for the quotient orienter is motivated by the observation that the quotient orienter succeeds on most rotations and fails on small subsets of rotations; see our response to Reviewer 3giS for a discussion of how this may originate from discontinuities in its outputs. Given an arbitrarily-oriented input shape $RS$, we would like to mitigate the possibility that the shape is in an orientation for which the quotient orienter fails. To do so, we apply $K$ random rotations $R_k$ to the input shape to obtain randomly re-rotated shapes $R_kRS$. Because the quotient orienter only fails on small subsets of rotations, we expect it to succeed on most of the $R_kRS$ and recover the orientations of $R_kRS$ up to an octahedral symmetry.
>
> Because we are interested in the orientation of $RS$ rather than $R_kRS$, we then need to apply the inverse of $R_k$ to each predicted orientation $f_\theta(R_kRS)$ to obtain a set of $K$ predicted orientations $R_k^\top f_\theta(R_kRS)$. These will be correct orientations of $RS$ up to octahedral symmetries for each $k$ where the quotient orienter succeeds.
>
> Because we expect this to be the case for most $k$, we use a voting scheme to choose one of the candidate orientations $R_k^\top f_\theta(R_kRS)$. (This is step (4) in B.1, which you have inquired about.) In this step, we compute each candidate’s average quotient $L_2$ loss w.r.t. every other candidate (i.e. the loss in Problem (3)), and choose the candidate which minimizes this measure. Because we expect the quotient orienter to have succeeded for most of the $R_kRS$, this average quotient loss will be small for most candidates and large for the few outlier candidates on which the orienter failed. Choosing the candidate with the minimum average quotient loss wrt the other candidates filters out these outliers and makes it likelier that we output the correct orientation of $RS$ up to an octahedral symmetry.
>
> **Flipper.**
>
> The TTA procedure is similar here. We are given an input shape $FS$, which is correctly-oriented up to an octahedral symmetry (a “flip”) $F$ if the quotient orienter succeeded. We apply $K$ random flips $F_k$ to the input shape to obtain randomly re-rotated shapes $F_kFS$ and apply the flipper to each of these shapes. Similarly to the previous case, if the flipper succeeds on some $F_kFS$, it predicts $F_kF$ rather than the flip $F$ we are actually interested in. We consequently left-multiply each prediction $g(F_kFS)$ by $F_k^\top$, which maps each successful prediction $g(F_kFS) = F_kF$ to the true flip $F$. Because we expect the flipper to have succeeded on most inputs $F_kFS$ and failed on a minority of inputs, we again use a voting scheme to pick out the plurality prediction. In this case, we simply return the most common flip among the set of $F_k^\top g(F_kFS)$.
>
> We do not use TTA when outputting adaptive prediction sets. We would be pleased to add these details to the relevant appendices in the camera-ready.
>
> **Proof of Prop. 3.1.**
>
> We now clarify the first step of the proof of Prop. 3.1. Because we do not place any restrictions on the functions $f : \mathcal{S} \rightarrow SO(3)$ other than being functions (i.e. not one-to-many), Problem (2) decouples over the inputs to $f$. That is, we can independently solve for the optimal $f^*(RS)$ for any input shape of the form $RS$ with $R \in SO(3)$. If $S$ has rotational symmetries, there will be several $R’ \in SO(3)$ for which $RS = R’S$. Because $f^*$ cannot be one-to-many, we must have $f^*(RS) = f^*(R’S)$ for all such $R’$. Consequently, the terms in Problem (2) that are relevant to determining $f^*(RS)$ are precisely those involving the $R’$ s.t. $RS = R’S$.
>
> This means that the $f^*$ which solves Problem (2) is defined at any input shape $RS$ as the rotation $f^*(RS) := R^* \in SO(3)$ which solves the problem on line 576.

---

> > ### Comment · Reviewer_gQb9 · 2025-04-02
> >
> > I carefully read all reviews and responses. The authors clarified how TTA is implemented and performed an ablation study on its contribution, which was found to be not critical for the overall performance of the method and its superior performance with respect to the baseline. The explanation of the first step of Prop 3.1 is also clear now. I'd suggest to add this explanation in the appendix. The other reviews do not uncover critical weaknesses and the responses to them seems convincing. For all these reasons, I confirm my overall recommendation. However, I agree that the title should be toned down, if possible. In particular, I'm in favor of the proposal by the authors "3D Orientation Estimation for Symmetric Shapes" (dropping "Robust" for the reasons discussed by reviewer ffTb).

---

### Official Review · Reviewer_3giS · 2025-03-12

**Overall Recommendation:** 4

**Summary:**

The paper introduces a two-stage method for estimating the pose of an object (3D point cloud). In the first stage, the pose is regressed modulo octahedral symmetries, which prevents prediction collapse for symmetric objects. In the second stage, the remaining octahedral ambiguity is resolved through classification. Experiments on ShapeNet/ModelNet show that the approach works well.

## update after rebuttal
The authors adequately responded to my comments. A refined title, refined propositions, and including the shown discontinuity plot in the paper/appendix improve the work. So, I have raised my score to 4.

**Claims And Evidence:**

Broadly, the claims are well supported and the paper is well written.

1. The conclusion states "*Whereas previous approaches can only
infer upright orientations for limited classes of shapes, our
method successfully recovers entire orientations for general
shapes.*" But, as mentioned in the introduction "*Our work may be viewed as
an efficient canonicalization method for the specific case of
3D shapes with rotational symmetries.*" In particular, earlier energy-based canonicalization approaches would not have an issue with symmetric objects.

2. This is slightly subjective, but the title could be more specific than "Orient Anything". In this paper, 3D point clouds of man-made objects are oriented.

**Essential References Not Discussed:**

None that I am aware of.

**Experimental Designs Or Analyses:**

1. Why is the network trained for "3719 epochs"? Is it early stopping with a larger number of epochs?
2. The inference time should be reported.
3. The proposed method uses TTA, does the baseline Upright-Net?

**Methods And Evaluation Criteria:**

The evaluation makes sense.

**Other Comments Or Suggestions:**

N/A

**Other Strengths And Weaknesses:**

- The authors state that the octahedral group is among the largest subgroups of SO(3), without mentioning the icosahedral group. It would be interesting to see if using the icosahedral group leads to worse performance due to not being aligned with the symmetries of common man-made objects.

**Questions For Authors:**

Let me copy my questions to this section for the convenience of the authors.

### Proposition 3.2
1. If I'm not mistaken, the proof shows that there exists an $f^*$ of the specified form, not that all minimizers are of the specified form (as claimed in the prop.).
2. Here is an example, which I think should be explained.
Let $S$ be a cube. Let $\hat{R}$ be the octahedral group. Consider rotating the cube around one of its symmetry axes continuously from 0 to 90 degrees. At 0 and 90 degrees, the output must be the same since the input is the same. Is there a discontinuity somewhere in between? What happens if one inputs a cube into the trained network and varies its rotation?
3. What happens when $S$ has more symmetries than present in $\hat{R}$? For instance, in the experiments, there are several objects that have SO(2) symmetry (e.g. vases). Does the regression objective degenerate for these, similarly to in Prop 3.1?

### Experiments
1. Why is the network trained for "3719 epochs"? Is it early stopping with a larger number of epochs?
2. What is the runtime of the proposed method and Upright-Net respectively?
3. The proposed method uses TTA, does the baseline Upright-Net?

I will raise my score to accept if these are reasonably addressed and no other important issues are discovered during the reviewing.

**Relation To Broader Scientific Literature:**

It is a promising idea to factor out symmetries in canonicalization methods. It requires the existence of a subgroup that covers the symmetries of standard inputs.

**Theoretical Claims:**

I skimmed most of the proofs. I checked Prop 3.2 in more detail.

In general, there is an issue where the minimization problems specified minimize over functions $f$, but it is not specified what function class $f$ belongs to. This can lead to problems, e.g., if the proof requires $f$ to be discontinuous, not aligning well with neural network approximators.

### Proposition 3.2
1. If I'm not mistaken, the proof shows that there exists an $f^*$ of the specified form, not that all minimizers are of the specified form (as claimed in the prop.).
2. Here is an example, which I think should be explained.
Let $S$ be a cube. Let $\hat{R}$ be the octahedral group. Consider rotating the cube around one of its symmetry axes continuously from 0 to 90 degrees. At 0 and 90 degrees, the output must be the same since the input is the same. Is there a discontinuity somewhere in between? What happens if one inputs a cube into the trained network and varies its rotation?
3. What happens when $S$ has more symmetries than present in $\hat{R}$? For instance, in the experiments, there are several objects that have SO(2) symmetry (e.g. vases). Does the regression objective degenerate for these, similarly to in Prop 3.1?

---

> ### Author Rebuttal · Authors · 2025-03-31
>
> Thank you for your thoughtful review and your helpful comments. We address your questions below.
>
> **Proposition 3.2:**
>
> *If I'm not mistaken, the proof shows that there exists an $f^\*$ of the specified form, not that all minimizers are of the specified form (as claimed in the prop.).*
>
> This is correct; thank you for pointing this out. We will gladly amend Prop. 3.2 in the camera-ready to state that there exists a solution of the specified form.
>
> *Let $S$ be a cube. Let $\hat{R}$ be the octahedral group. Consider rotating the cube around one of its symmetry axes continuously from 0 to 90 degrees. [...] Is there a discontinuity somewhere in between? What happens if one inputs a cube into the trained network and varies its rotation?*
>
> In practice, our trained orienter does exhibit discontinuities at certain rotations. To demonstrate this, we perform an experiment on a bench shape, which is symmetric under a 180 degree rotation about the y-axis. We rotate the bench around this axis in increments of 1 degree and pass the resulting shape into our orienter for each rotation. We track two metrics throughout this process:
>
> - The quotient loss of our orienter’s prediction (i.e. the loss function from problem (3), where we quotient the $L_2$ loss by the octahedral group). This measures the accuracy of our orienter’s predictions up to an octahedral symmetry.
> - The chamfer distance between the shapes obtained by applying the orienter’s output to the rotated bench at subsequent angles of rotation. This detects discontinuities in the orienter’s output.
>
> We have plotted both of these metrics across the rotation angle [here](https://imgur.com/a/hjvfOM1). Our orienter exhibits several sharp discontinuities, manifested as spikes in the chamfer distance plot. These discontinuities are associated with spikes in the quotient loss, which are occasionally large. However, these spikes in the quotient loss are highly localized, and our orienter performs well for the vast majority of rotations. These localized spikes motivate our use of TTA to improve our pipeline’s performance. We provide further details on TTA in our response to Reviewer gQb9.
>
> *What happens when $S$ has more symmetries than present in $\hat{R}$? For instance, in the experiments, there are several objects that have SO(2) symmetry (e.g. vases). Does the regression objective degenerate for these, similarly to in Prop 3.1?*
>
> If S has more symmetries than $\hat{R}$, then we would expect the regression objective to degenerate. As you note, this typically occurs for shapes with continuous symmetries such as vases, because the octahedral group covers many of the symmetries of shapes with finitely-many rotational symmetries. However, this degeneracy is in fact benign for such shapes. For example, the solution to (2) for a vase which is symmetric under any rotation about its up-axis may be any rotation about the up-axis (the shape’s continuous axis of symmetry). But unlike the bench shape that we discuss in lines 192-205, any such rotation is a valid orientation of the vase up to one of its symmetries, so our pipeline will output a rotation that returns the vase to its correct pose (up to a symmetry).
>
> **Experiments:**
>
> *Why is the network trained for "3719 epochs"? Is it early stopping with a larger number of epochs?*
>
> We were primarily bottlenecked by computational constraints, and the validation accuracy was continuing to increase (albeit very slowly) when we decided to end our final training run. We believe that our pipeline’s accuracy could be slightly improved with further training, particularly for the flipper.
>
> *What is the runtime of the proposed method and Upright-Net respectively?*
>
> Our method’s inference time with TTA is 0.4517 seconds/shape (std=0.0252). Its inference time without TTA is 0.0126 seconds/shape (std=0.0466). Upright-Net’s inference time is 0.0672 seconds/shape (std=0.0252). We will add these details to the camera-ready.
>
> Our method’s inference time without TTA is faster than Upright-Net’s, but employing TTA makes our method’s inference time notably longer than Upright-Net’s. However, the new metrics that we have computed for our pipeline without TTA (links below) show that our method continues to strongly outperform Upright-Net if we omit TTA.
>
> *The proposed method uses TTA, does the baseline Upright-Net?*
>
> Upright-Net does not use TTA. For the sake of comparison, we have re-run the experiments in our paper without TTA. The angular error plots in Figure 6, now including our method without TTA, are [here (6a, Shapenet)](https://imgur.com/a/YpsbHrw) and here [(6b, ModelNet40)](https://imgur.com/a/FwyEydw). A screenshot of a new version of [Table 1 is here](https://imgur.com/a/y5KYtOR), and [Table 2 is here](https://imgur.com/a/4jBy2ko).
>
> Omitting TTA causes our method’s up-axis estimation accuracy to fall by roughly 4-5% depending on the dataset, but our method continues to strongly outperform Upright-Net without TTA.

---

### Decision · Program_Chairs · 2025-05-01

**Decision:**

Accept (poster)

**Comment:**

This paper introduces Orient Anything, a two-stage framework for estimating the full 3D orientation of objects, particularly addressing challenges posed by symmetrical shapes. The first stage employs a quotient orienter that predicts the object's orientation up to the 24 rotational symmetries of the octahedral group, effectively handling ambiguities inherent in symmetric objects. The second stage utilizes a flipper to classify and resolve these symmetries, refining the orientation prediction to a unique solution. To manage residual uncertainties, especially in cases of inherent ambiguity, the method incorporates conformal prediction, providing adaptive prediction sets.

This paper was appreciated by all reviewers. Reviewer 3giS and gQb9 gave accept scores, and reviewer yNUC and ffTb gave weak accept scores. The authors provided extensive comments to clarify reviewers' concerns about technical details and experiment settings. AC recommends applying the reviewers' comments and suggestions in the revised version. Especially since the paper title is ambitious, the reviewer ffTb's final comment regarding testing with reconstructed 3D shapes by other approaches is reasonable.